# Ketamine Produces a Long-Lasting Enhancement of CA1 Neuron Excitability

**DOI:** 10.3390/ijms22158091

**Published:** 2021-07-28

**Authors:** Grace Jang, M. Bruce MacIver

**Affiliations:** Department of Anesthesiology, Perioperative and Pain Medicine, School of Medicine, Stanford University, Stanford, CA 94305, USA; gjang@stanford.edu

**Keywords:** ketamine, antidepressant, anesthetic, hippocampus, neuron excitability, NMDA, GABA

## Abstract

Ketamine is a clinical anesthetic and antidepressant. Although ketamine is a known NMDA receptor antagonist, the mechanisms contributing to antidepression are unclear. This present study examined the loci and duration of ketamine’s actions, and the involvement of NMDA receptors. Local field potentials were recorded from the CA1 region of mouse hippocampal slices. Ketamine was tested at antidepressant and anesthetic concentrations. Effects of NMDA receptor antagonists APV and MK-801, GABA receptor antagonist bicuculline, and a potassium channel blocker TEA were also studied. Ketamine decreased population spike amplitudes during application, but a long-lasting increase in amplitudes was seen during washout. Bicuculline reversed the acute effects of ketamine, but the washout increase was not altered. This long-term increase was statistically significant, sustained for >2 h, and involved postsynaptic mechanisms. A similar effect was produced by MK-801, but was only partially evident with APV, demonstrating the importance of the NMDA receptor ion channel block. TEA also produced a lasting excitability increase, indicating a possible involvement of potassium channel block. This is this first report of a long-lasting increase in excitability following ketamine exposure. These results support a growing literature that increased GABA inhibition contributes to ketamine anesthesia, while increased excitatory transmission contributes to its antidepressant effects.

## 1. Introduction

Ketamine is an anesthetic, utilized in clinical settings since 1970 [1,2]. It has been shown to produce amnestic, analgesic, and anti-inflammatory effects [2]. Ketamine has proven to be a fast-acting and long-lasting antidepressant, first demonstrated in patients with major depressive disorder in 2000 [3]. In contrast to traditional monoaminergic antidepressants, ketamine is known to depress N-methyl-D-aspartate (NMDA) signaling [4,5]. In addition to NMDA receptors, ketamine has been shown to act on γ-aminobutyric acid (GABA) receptors [2,6,7], muscarinic and nicotinic acetylcholine receptors [2,8,9], dopamine and serotonin receptors, opioid receptors, sigma receptors [2,9], hyperpolarization-activated cyclic nucleotide-gated channels [2,10,11], potassium channels [10,11,12,13,14], sodium channels, and calcium channels [2,12,15]. While ketamine’s anesthetic effects have been mostly attributed to its actions on NMDA and GABA receptors, there is a growing controversy as to whether the inhibition of NMDA receptors is involved in its antidepressant effects [2,4,5,16,17,18,19]. Some evidence suggests that ketamine’s antidepressant effects are due to a blockade of NMDA receptors producing a downregulation of eukaryotic elongation factor 2 (eEF2) kinase and the subsequent enhancement of brain-derived neurotrophic factor (BDNF) expression [4,5]. This results in an increased α-amino-3-hydroxy-5-methyl-4-isoxazolepropionic acid receptor insertion on the postsynaptic membrane leading to synaptic potentiation, protein synthesis and synaptogenesis [4,5,16]. Conversely, evidence also indicates actions independent of NMDA receptors, since ketamine’s metabolites have been shown to produce antidepressant effects without inhibiting NMDA receptors at clinically relevant concentrations [17,18,19]. The present study examined synaptic, neuronal and circuit-level loci of ketamine’s actions on glutamate-mediated synaptic transmission, the duration of these actions, and the involvement of NMDA receptors in these actions. The rationale for this study was to determine whether NMDA receptors were critical to ketamine’s action and whether long-lasting effects were produced. With its well-defined lamellar structure of glutamatergic and GABAergic synapses, the hippocampus provided a model to explore these effects of ketamine [20,21].

## 2. Results

### 2.1. Ketamine’s Acute Drug Effect: Decrease in Population Spike (PS) Amplitudes with an Enhancement of GABA-Mediated Inhibition

We tested concentrations of ketamine at 10 µM (antidepressant) [4,5] and 350 µM (anesthetic) [22] on PSs recorded from mouse hippocampal slices (Figure 1). At 10 µM, ketamine produced a small decrease in PS amplitudes (−2% ± 11%; *n* = 7) that increased back to, and beyond baseline during the latter half of drug application and continued into washout (Figure 1A,B). At 350 µM, ketamine produced a larger depression in PS amplitudes (−6% ± 8%; *n* = 6) that was sustained for the duration of drug application but was reversed upon washout (Figure 1C,D).

Application of a GABA receptor antagonist bicuculline (10 µM) reversed the depression in PS amplitudes produced by ketamine (350 µM) (Figure 2A,B). This indicates an involvement of GABA receptors in ketamine’s action. Additionally, ketamine (350 µM) increased the ratio of paired-pulse inhibition during application, and this was also reversed by bicuculline (10 µM) (Figure 2C), indicating a ketamine enhancement of GABA-mediated inhibition, as previously reported [6].

### 2.2. Ketamine’s Long-Lasting Effect: Increase in PS Amplitudes upon Washout Is Produced by a Post-Synaptic Mechanism

The long-term effects of ketamine can be seen during washout, as evidenced by a long-lasting increase of PS amplitudes and an increase in paired pulse inhibition. A ketamine amount of 10 µM produced an increase in PS amplitudes of 10% ± 7% at 60 min into washout, and this effect continued for the duration of recording, resulting in an increase of 15% ± 5% at 120 min into washout (Figure 1A,B). A ketamine amount of 350 µM produced an increase in PS amplitudes of 15% ± 4% at 60 min into washout, which increased to 16% ± 8% at 120 min into washout (Figure 1C,D). This washout effect was exhibited at the testing of both concentrations and was similar to long-term potentiation (LTP) [4,5], which is characterized by a rapid increase in amplitudes that is sustained for several hours. Because of its rapid onset (within minutes), we postulated that this effect was due to changes in synaptic strength during ketamine application, as previously shown by the Kavalali and Monteggia labs [4,5].

However, the mechanism behind the LTP-like effect for our responses appears to be postsynaptic rather than synaptic in origin. We tested this by studying the effects of ketamine (10 µM) on field excitatory postsynaptic potentials (fEPSPs) (Figure 3A,B). If the LTP-like effect was produced by conventional means, we would expect to see an increase in fEPSP amplitudes following the washout of ketamine, which would signify an increase in synaptic transmission that is causing this excitatory effect. However, unlike its effect on PS amplitudes, ketamine did not decrease fEPSP amplitudes during application but rather slightly increased them. Moreover, ketamine did not produce a significant increase in fEPSP amplitudes during washout. While previous studies have shown that ketamine could potentiate fEPSPs [4,5], our results indicate that this effect was not long-lasting.

To determine whether ketamine increased CA1 neuron excitability rather than synaptic strength, we compared the amplitude of PSs to the slope of fEPSPs before and after the 10 µM ketamine application (during washout). Stimulus strength was varied between 4–11 V to produce responses across a range of fEPSP slopes (input) and PS amplitudes (output). The resulting input-output curve has a steeper slope following ketamine washout compared to before ketamine application (Figure 3C). Ketamine (10 µM) increased the slope of the input-output curves from 39 ± 6 to 48 ± 12 (*n* = 3). These results indicate that ketamine produces an increased excitability of CA1 neurons, with little effect on fEPSP input.

### 2.3. Mechanisms behind Ketamine’s Long-Lasting Effect May Involve NMDA Receptor Antagonism

LTP is commonly described as an NMDA receptor-dependent phenomenon [23], and ketamine has been shown to block LTP [24]. Furthermore, the mechanisms behind ketamine’s anesthetic effect have been ascribed to NMDA receptor antagonism [2,25]. In line with this evidence, we wanted to test whether ketamine’s ability to produce a long-lasting increase in PS amplitudes was due to NMDA receptor antagonism. We examined this using a more selective NMDA receptor antagonist, DL-2-Amino-5-phosphonovaleric acid (APV) (Figure 4A,B). We have previously shown that APV (100 µM) completely blocks NMDA receptor-mediated fEPSPs [26]. Like ketamine, APV produced a small decrease in PS amplitudes during application (−1% ± 4%; *n* = 10) which was reversed and increased upon washout. However, the increase during APV washout was much smaller in magnitude (4% ± 5% at 60 min into washout) and not sustained long-term, with amplitudes returning to baseline by 110 min into washout.

We then tested MK-801, which blocks the calcium permeable ion channel of the NMDA receptor complex [27]. MK-801 was able to reproduce both the acute and long-term effects of ketamine (Figure 4C,D). During application, MK-801 produced a small decrease in PS amplitudes (−1% ± 2%; *n* = 7) which was reversed upon washout. Like ketamine, MK-801 produced a robust increase in PS amplitudes during washout (12% ± 8% at 60 min into washout), which continued for the duration of the recording (15% ± 9% at 120 min into washout). These results indicate that ketamine shares similar actions with MK-801 for producing long-lasting effects. The differences between the results produced by ketamine and MK-801 compared to APV highlight the importance of differentiating the location at which NMDA signaling is blocked (receptor vs. calcium ion channel).

Since ketamine has also been shown to inhibit potassium channels [10,11,12,13,14], we tested tetraethylammonium (TEA), a broad-spectrum potassium channel blocker, to examine whether potassium channel block could contribute to ketamine’s effects (Figure 4E,F). During application, TEA (25 mM:[28]) produced a large decrease in PS amplitudes (−74% ± 24%; *n* = 9), which was much more pronounced than that produced by ketamine (350 µM). Like ketamine, TEA produced a long-lasting increase in PS amplitudes during washout (12% ± 7% at 60 min into washout), which was sustained for two hours (8% ± 8% at 120 min into washout). This indicates a potential involvement of potassium channels in ketamine’s long-lasting effect.

As graphically summarized in Figure 5, ketamine consistently produced decreased PS amplitudes during application at both antidepressant and an anesthetic concentrations. Although this effect did not achieve statistical significance at either concentration tested, the trend was clearly evident in Figure 1. Upon washout, ketamine was able to produce statistically significant, long-lasting increases in PS amplitudes at 60 min (*p* < 0.01, Wilcoxon Rank Test) and 120 min into washout (*p* < 0.01, Wilcoxon Rank Test) at both concentrations. While APV (100 µM) produced an increase in PS amplitudes at 60 min into washout, this effect was neither statistically significant nor long-lasting as mean amplitudes had decreased back to baseline by 120 min into washout. Interestingly, in the presence of APV (100 µM), ketamine (10 µM) was still able to increase PS amplitudes during washout, although this effect was not statistically significant. MK-801 (40 µM) was able to reproduce ketamine’s effects by decreasing PS amplitudes during application and sustaining a statistically significant increase in PS amplitudes at 60 min (*p* < 0.01, Wilcoxon Rank Test) and 120 min into washout (*p* < 0.01, Wilcoxon Rank Test). TEA was the only tested-drug that produced a statistically significant decrease in PS amplitudes during application (*p* < 0.01, Wilcoxon Rank Test), which likely resulted from depolarization block of CA1 neurons [29]. During washout, TEA was able to produce a statistically significant increase in PS amplitudes at 60 min (*p* < 0.01, Wilcoxon Rank Test) and 120 min into washout (*p* < 0.05, Wilcoxon Rank Test).

## 3. Discussion

Previous work from our lab has shown that other anesthetics depress PS amplitudes during application [30], but ketamine is the only anesthetic that has ever been shown to produce a long-lasting increase in PS amplitudes during washout. Ketamine’s ability to increase fEPSP amplitudes during washout has been reported by the Kavalali and Monteggia labs [4,5]; however, the effect was only recorded for one hour into washout. Ours is the first demonstration of a long-lasting increase in excitability following washout of ketamine from brain slices. An increase in PS amplitudes following washout was also seen with MK-801, but was only partially evident with APV, demonstrating the importance of ion channel blocking downstream of NMDA receptors. Additionally, our results for TEA indicate a potential involvement of potassium channel blocking in ketamine’s long-lasting effect. The main findings from the study include ketamine’s ability to produce an acute depression in PS amplitudes by increasing GABA-mediated inhibition [6] during application (Figure 2) and a rapid, long-term enhancement of PS amplitudes during washout. This long-lasting effect was statistically significant and produced by a postsynaptic mechanism, demonstrated by a lack of effect on fEPSPs (Figure 3A,B) and by the upward shift in input-output curves (Figure 3C). We differentiate postsynaptic effects from presynaptic effects by comparing ketamine’s action on fEPSPs with those on CA1 neuron population spikes (PS). We saw no change in fEPSPs when ketamine was applied nor was there any evidence of a post-washout increase; this can be interpreted as little to no effect at the synapse as shown in Figure 3A. We also performed an input-output analysis (Figure 3C), which showed that PS amplitudes increased with little to no change in fEPSP responses. Thus, ketamine is acting downstream from the synapse to increase the discharge of CA1 neurons (PS).

Our results suggest that both NMDA- and non-NMDA-mediated actions may contribute to ketamine’s effects. The variation in results from our ketamine, MK-801, and APV experiments highlight the differences in inhibiting at various locations of the protein signaling complex which includes NMDA receptors, G-protein-coupling, and calcium ion channels. We speculate from the results of our MK-801 experiments that by inhibiting the calcium ion channels linked to NMDA receptors, ketamine may indirectly change levels of intracellular calcium [31] and increase intrinsic postsynaptic neuronal excitability by lowering the activation threshold of pyramidal neurons [11]. These changes could activate any number of downstream signaling pathways to produce long-lasting structural changes, including those involving calcium-dependent protein kinase II, C, or A [23]. As previously proposed, another plausible pathway could involve eEF2 kinase, BDNF, and rapamycin or glycogen synthase kinase 3 [17,32]. Yet another possibility could involve the NMDA receptor coagonist D-serine and its relationship to ketamine’s long-lasting effects. Both ketamine and MK-801 have been shown to elevate D-serine levels which could influence synaptic plasticity via indirect modulation of NMDA and GABA receptors [33,34].

In addition to NMDA receptors, other mechanisms such as potassium channels could contribute long-term increases in excitability [2,35]. This is supported by the results from our TEA experiments that also produced a long-lasting effect and replicates previous findings showing that ketamine blocks calcium-activated potassium currents [36]. Moreover, ketamine’s acute drug effects appear to involve an enhancement of GABA-A inhibition, as reported by the Orser Lab [6]. An increase in GABA-mediated inhibition would account for the decrease in PS amplitudes and the increase in paired-pulse inhibition as seen during the application of ketamine in our experiments [6]. This is supported by bicuculline’s ability to reverse ketamine’s acute effects.

There are several limitations to our study that can provide directions for future research. We used APV and MK-801 to examine the role of NMDA receptors in producing ketamine’s effects. Future studies using more selective NMDA receptor antagonists will be needed to identify the specific types of NMDA receptors, the associated ion channels, and the location on the NMDA receptor complex that ketamine acts. Similarly, we tested TEA, a broad-spectrum potassium channel blocker, to demonstrate the general involvement of potassium channels in the long-lasting effects produced by ketamine. Further research utilizing more specific potassium channel blockers are needed to determine specific potassium channels for ketamine’s actions. Additionally, while we used bicuculline to examine the involvement of GABA receptors during a ketamine application [6], future studies using other GABA receptor antagonists such as picrotoxin will be needed to verify the role of GABA receptors versus chloride channels in ketamine’s effects. Furthermore, only male mice were used in this study to avoid complicating results [37,38], but additional studies using female rodents would be beneficial to explore differences in ketamine actions between the sexes [39]. Future studies using whole-cell patch-clamp recording methods will be needed to measure changes in CA1 neuron excitability in response to ketamine.

We recognize that findings from an in vitro mouse brain slice preparation are not likely to capture the complexities involved in the brain pathology of depressed patients; however, a lasting increase of pyramidal neuron excitability (in hippocampus, neocortex and elsewhere) provides a plausible mechanism for ketamine’s antidepressant effects in vivo [40]. Of course, hippocampal glutamate and GABA synapses may not be representative of the brain areas commonly associated with the production and regulation of emotions ;therefore, prefrontal cortex and other limbic regions may be better for future studies [41]. Other limitations include the lack of information regarding the effect site concentration of ketamine in human subjects from clinical research studies, as well as the unknown concentration of ketamine within hippocampal slices at the site of action in our experiments [42]. Lastly, since our experiments were performed at room temperature, the results may not be indicative of the effects of ketamine at physiological temperatures [43].

Nonetheless, the rapid onset and sustained effects of ketamine as demonstrated in the present study do resemble the time course of ketamine’s rapid clinical antidepressant effect. New research suggests that antidepressant effects occur at both low concentrations of ketamine for the clinical treatment of depression and high concentrations of ketamine used during general anesthesia (personal communication from Boris D. Heifets, based on his new clinical trial results). We interpret our results as evidence that ketamine acts downstream from NMDA receptors to produce a long-lasting increase in CA1 neuron excitability that could contribute to its antidepressant effects by shifting the excitation/inhibition coupling towards a more excitable state.

## 4. Materials and Methods

### 4.1. Brain Slice Preparation 

All experimental protocols were approved by the Institutional Animal Care and Use Committee and the Administrative Panel on Laboratory Animal Care at Stanford University. Methods for brain slice preparation and electrophysiological recordings have been described elsewhere [30], but we include a cursory description here. Brain slices were obtained from male C57BL/6J mice weighing between 25–30 g from The Jackson Laboratory (Bar Harbor, ME, USA). Previous studies have shown sex differences in ketamine’s effects [37,38], so female mice were not used to avoid complicating results. Mice were deeply anesthetized with isoflurane carried by carbogen (95% oxygen and 5% carbon dioxide) using an Isotec 3 vaporizer (Datex-Ohmeda, Chicago, IL, USA) and euthanized by decapitation. Once removed, brains were placed in ice cold, oxygenated artificial cerebrospinal fluid (ACSF). ACSF composition included the following in OmniSolv water (Millipore Sigma, Burlington, MA, USA): 124 NaCl, 26 NaHCO_3_, 3.5 KCl, 2 MgSO_4_, 2 CaCl_2_, 1.25 NaH_2_PO_4_, and 10 D-glucose (mM concentrations). Within a minute of placement in the ACSF solution, the brain was hemisected along its sagittal plane. Brain halves were placed back into the oxygenated, ice cold ACSF solution, and each hemisphere was individually sectioned into 400 µM thick coronal slices with a McIlwain Tissue Chopper (Cavey Laboratory Engineering Co. Ltd., Gomshall, UK). Slices were placed on cellulose filter papers, transferred into a humidified incubation chamber with liquid ACSF bubbling with carbogen at 30 °C for one hour and equilibrated at room temperature (22–24 °C) for at least another hour. Individual slices were submerged in a recording chamber perfused with oxygenated, room temperature ACSF at a flow rate of 3.0 mL/min for electrophysiological recordings.

### 4.2. Electrophysiological Recordings

Extracellular evoked field potentials—population spikes (PSs) and field excitatory postsynaptic potentials (fEPSPs)—were recorded (Figure 6). Bipolar tungsten stimulating microelectrodes (Frederick Haer & Co., Bowdoin, ME, USA) were placed in the CA1 stratum radiatum of the hippocampus to stimulate Schaffer collateral axons using a Grass S8800 stimulator (Grass Instrument Co., Quincy, MA, USA). Recording electrodes were made from glass micropipettes (Garner Glass, Claremont, CA, USA) that were pulled with a two-stage pipette puller (Model PP-830, Narishige, Japan) to a 2–5 MOhm tip resistance, when filled with ACSF. Recording electrodes were placed near the CA1 cell body layer in the stratum oriens for PS responses or in the stratum radiatum for fEPSP responses. Extracellular recordings of PS and fEPSP responses were used throughout this study. Paired-pulse stimulation with a 100 ms interpulse interval was delivered every 20 s at a stimulus intensity of 4–10 V. Field potentials were recorded with an Axoclamp-2A amplifier (Axon Instruments, Foster City, CA, USA), digitized and stored using Igor Pro (Wavemetrics, Inc., Oswego, OR, USA) at a sampling rate of 20 kHz. Stable baseline responses (<2% variation) were recorded and displayed in real time for 20 min prior to drug application for experiments. Concentrations of compounds tested include 10 µM and 350 µM ketamine, 100 µM DL-2-Amino-5-phosphonovaleric acid (APV), 40 µM MK-801, 10 µM bicuculline, and 25 mM tetraethylammonium (TEA) in a vehicle of oxygenated, control ACSF. All drugs were obtained from Sigma-Aldrich (St. Louis, MO, USA).

### 4.3. Statistical Analysis

Data analyses were performed using Igor Pro software. Combined experiment graphs were constructed from data sets comprised of several, averaged, time-matched experiments and displayed as mean amplitudes ± standard deviation versus experimental time. Amplitudes were normalized and expressed as a percent of control recordings taken in the 20 min of baseline prior to the drug application. For input-output curves, fEPSP slopes were measured between 10% and 90% of peak amplitudes, while PS amplitudes were measured as a minimum to peak positive response. Paired-pulse inhibition (PPI) was measured as a ratio of the amplitudes of the first PS divided by the second PS. Statistical analyses used the Wilcoxon Rank Test to compare control conditions to each experimental condition during the drug application, washout at 60 min and washout at 120 min. Statistical significance was set at *p* < 0.05. Each experimental condition consisted of four to ten experiments that were repeated using slices from different animals.

## Figures and Tables

**Figure 1 ijms-22-08091-f001:**
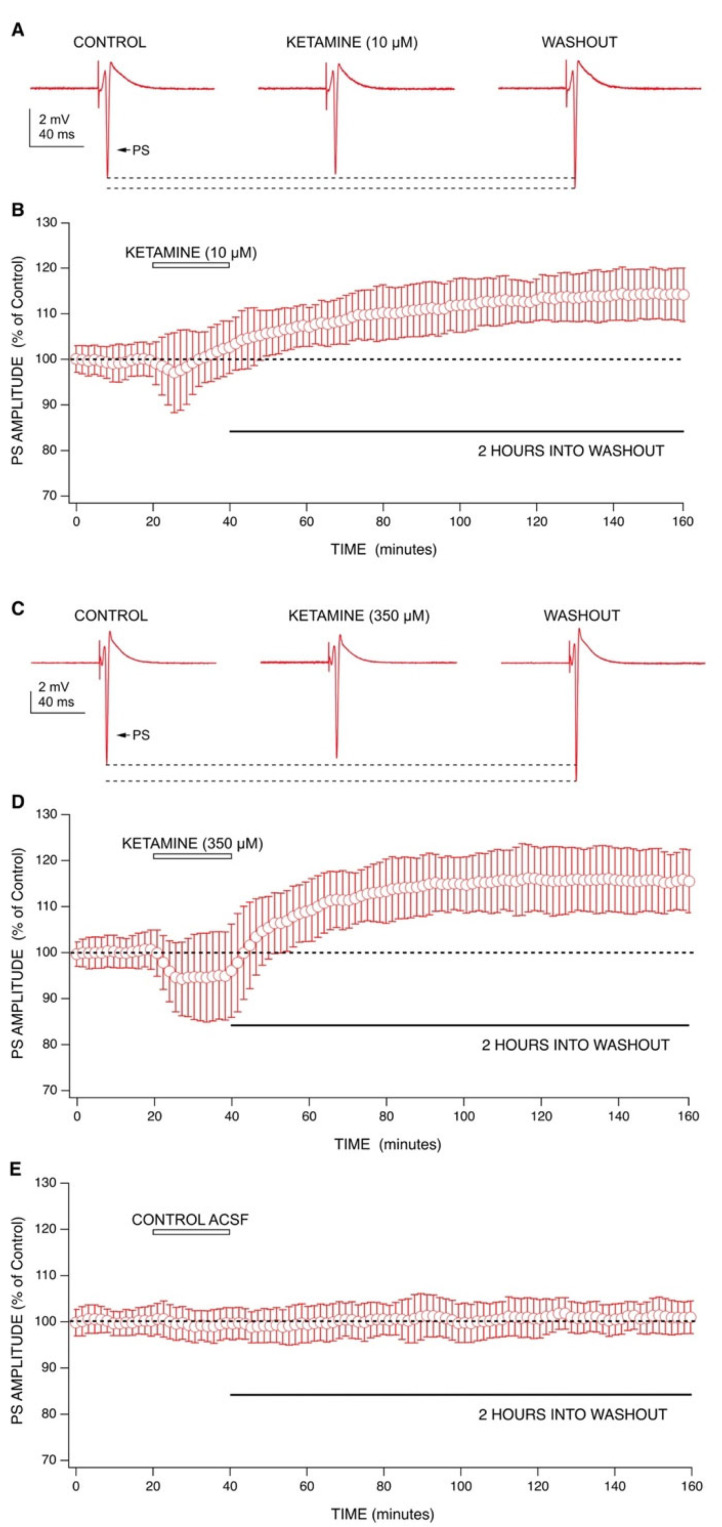
The effects of ketamine on population spike amplitudes. (**A**) Representative recordings from one ketamine experiment (10 µM) showing changes in PS amplitudes during application and washout. (**B**) Combined graph of PS amplitudes vs. experimental time (*n* = 7) shows 10 µM ketamine produced a small depression in PS amplitudes during the first 10 min of perfusion. During the latter half of the application, ketamine appeared to have an increasing effect on PS amplitudes, which continued into washout. Amplitudes increased beyond baseline during washout and persisted for two hours. (**C**) Representative recordings from one ketamine experiment (350 µM) showing changes in PS amplitudes during application and washout. (**D**) Combined graph (*n* = 6) shows 350 µM ketamine produced a depression in PS amplitudes during application. Conversely, an increase in PS amplitudes beyond baseline was seen during washout of ketamine which lasted for two hours. Comparing the two concentrations studied, there were no differences seen in the magnitudes of post-washout increases in excitability. (**E**) Combined graph (*n* = 7) shows no change in population spike amplitudes when switching between two control ACSF solutions. Data points from each combined graph represent the mean ± standard deviation derived from a data set of several averaged experiments decimated from approximately 480 to 100 data points.

**Figure 2 ijms-22-08091-f002:**
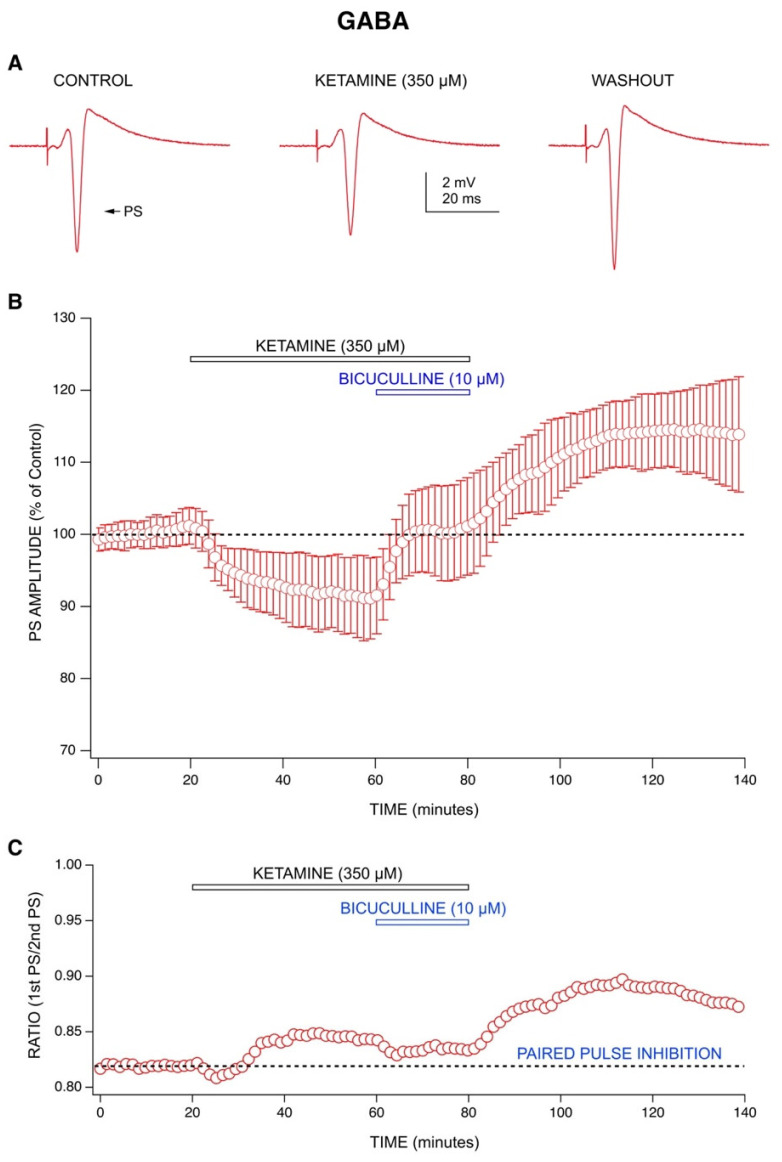
Evaluating the mechanisms behind ketamine’s effects: GABA receptors. (**A**) Representative recordings from one ketamine experiment (350 µM) showing changes in PS amplitudes during application and washout. (**B**) Combined graph (*n* = 5) shows 350 µM ketamine produced a depression in PS amplitudes during application which was reversed by 10 µM bicuculline. An increase in PS amplitudes was evident upon washout of ketamine and bicuculline. Each data point represents the mean ± standard deviation derived from a data set of five averaged experiments decimated from approximately 420 to 100 data points. (**C**) Paired-pulse inhibition (PPI), calculated as a ratio of the amplitude of the first population spike (PS) divided by the second, was increased in the presence of 350 µM ketamine (*n* = 5). This effect was partially reversed by 10 µM bicuculline. During washout of both ketamine and bicuculline, there was an increase in PPI.

**Figure 3 ijms-22-08091-f003:**
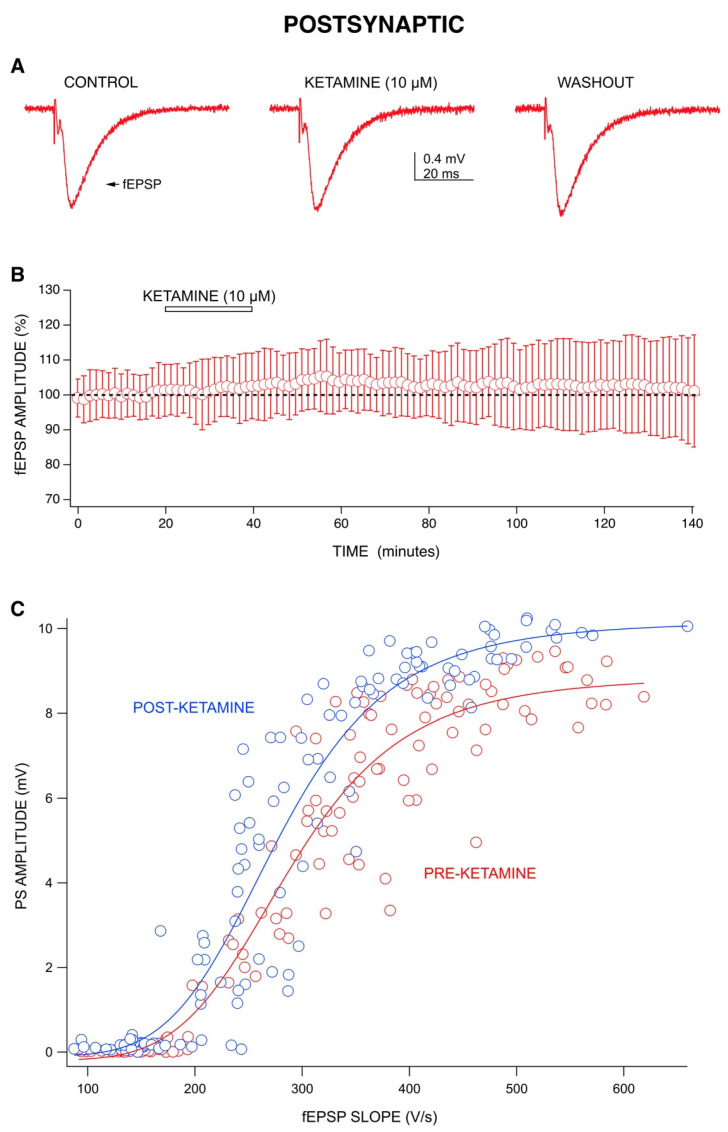
Evaluating the mechanisms behind ketamine’s effects: Postsynaptic actions. (**A**) Representative recordings from one ketamine experiment (10 µM) showing little change in fEPSP amplitudes during application and washout. (**B**) Combined graph (*n* = 9) shows ketamine (10 µM) did not have an effect on fEPSP amplitudes. Each data point represents the mean ± standard deviation derived from a data set of nine averaged experiments decimated from approximately 420 to 100 data points. (**C**) The relation of fEPSP slopes to PS amplitudes, with increasing stimulus intensities, revealed that a steeper slope was observed for post-ketamine responses. A quantity of 10 µM ketamine increased the amplitudes of PSs but not the slopes of fEPSPs. Data were fit to a Hill equation using Igor Pro.

**Figure 4 ijms-22-08091-f004:**
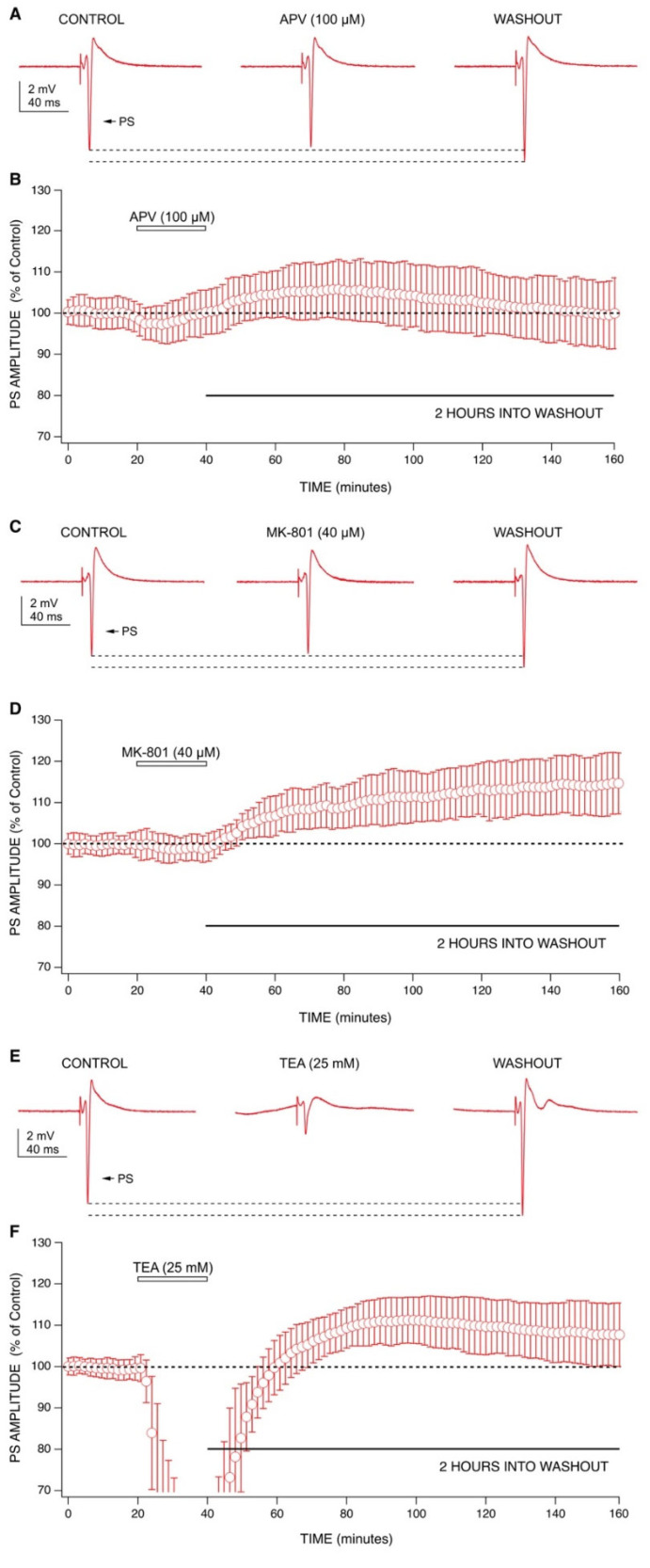
The effects of APV, MK-801, and TEA on population spike amplitudes. (**A**) Representative recordings from one APV experiment (100 µM) showing changes in PS amplitudes during application and washout. (**B**) Combined graph (*n* = 10) shows 100 µM APV produced a small depression in PS amplitudes during application. PS amplitudes increased during washout of APV, lasting approximately 40 min before decreasing back to baseline by 150 min (110 min into washout). (**C**) Representative recordings from one MK-801 experiment (40 µM) showing changes in PS amplitudes during application and washout. (**D**) Combined graph (*n* = 7) shows 40 µM MK-801 produced a small decrease in PS amplitudes during application. Washout of MK-801 produced a sustained increase in PS amplitudes for two hours. (**E**) Representative recordings from one TEA experiment (25 mM) showing changes in PS amplitudes during application and washout. (**F**) Combined graph (*n* = 9) shows 25 mM TEA produced a large depression in PS amplitudes during application. Washout of TEA produced an increase in PS amplitudes that was sustained for two hours. Data points from each combined graph represent the mean ± standard deviation derived from a data set of several averaged experiments decimated from approximately 480 to 100 data points.

**Figure 5 ijms-22-08091-f005:**
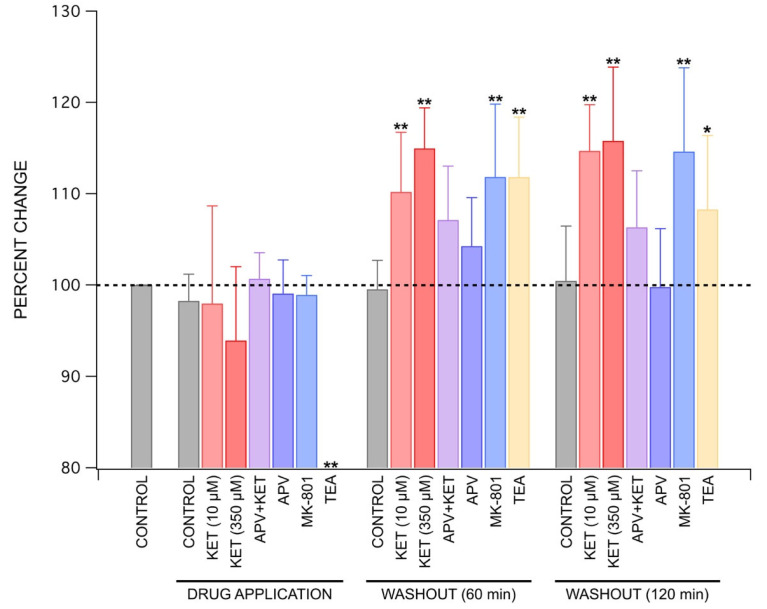
Summary of drug effects on population spike amplitudes. Ketamine decreased PS amplitudes during application at both anesthetic and antidepressant concentrations, with 350 µM exhibiting a larger decrease compared to 10 µM. Both concentrations of ketamine significantly increased PS amplitudes at 60 min and 120 min into washout, with 350 µM inducing the larger increase of the two. MK-801 was able to best replicate the effects produced by ketamine, as evidenced by a decrease in PS amplitude during application, followed by a statistically significant increase in amplitudes during washout at both 60 and 120 min. Similarly, washout of TEA was able to produce a statistically significant increase in PS amplitudes at both 60 min and 120 min. However, the decrease in PS amplitudes produced during application of TEA was much larger than that produced by ketamine, and statistically significant. Each bar represents the mean ± standard deviation for at least four experiments using separate brain slices prepared from different animals. Statistically significant differences are indicated using ** (*p* < 0.01), * (*p* < 0.05). Significance was assessed using Wilcoxon Rank Test to compare each experimental condition to its respective control.

**Figure 6 ijms-22-08091-f006:**
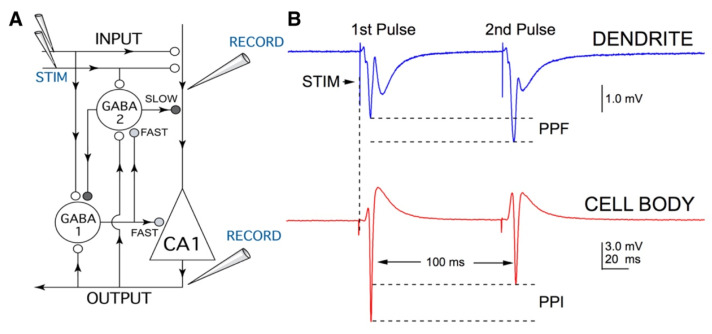
Diagram of inhibitory neuronal circuitry in the hippocampal CA1 region. (**A**) Schematic diagram showing a CA1 pyramidal neuron (triangle) and γ-aminobutyric acid (GABA)-mediated inhibitory interneurons (large circles). Glutamatergic synapses are represented using small open circles while GABAergic synapses are illustrated using shaded circles (light for ‘fast’ and dark for ‘slow’). The direction of information flow in axons and dendrites is indicated by arrows. For field excitatory postsynaptic potentials (fEPSPs), the recording electrode (RECORD) was placed in the apical dendrite region of the stratum radiatum. For population spikes (PS), the recording electrode was positioned along the axonal output side of the cell body layer of CA1 pyramidal cells (OUTPUT). Both fEPSP and PS responses were generated by stimulation with bipolar tungsten microelectrodes (STIM) placed on Schaffer collateral axons (INPUT). A paired-pulse stimulation paradigm with a 100 ms interpulse interval was used to assess the strength of GABA-mediated inhibition. (**B**) On the top is a recording showing paired-pulse facilitation (PPF) of fEPSPs, which results from increased excitatory glutamate release following the second stimulus pulse. A simultaneous recording on the bottom shows paired-pulse inhibition (PPI) of PSs, which results from enhanced inhibitory GABA transmission for the second pulse.

## Data Availability

Not Applicable.

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
