# Peer review of "Ketamine Produces a Long-Lasting Enhancement of CA1 Neuron Excitability"

_ijms, 2021, doi:10.3390/ijms22158091_

Round 1

Reviewer 1 Report

In this work, the investigators attempted to study effect of ketamine on long-lasting enhancement of CA1 neuronal excitability. Some of major comments regarding the experiments are shown in the following.

  • In Figures 1A, 1B, 1C and 1D, the concentration of ketamine was noticeably used at 10 and 350 microM. However, the PS amplitude (% of control) appeared no difference between these two concentrations. Please show the “dose-dependent” effect on changes in PS amplitudes.
  • In Figures 1C, D and Figure 2 (and lines 331-332), the concentration of ketamine used is 350 microM, a value which is surprisingly higher and difficult to be reached in the hippocampal region occurring in vivo. In lines 290-292, the statement was hence over-stated. Is there any pharmacokinetic studies which can be quoted or cited for this study? The typical antidepressant dosage of ketamine used to treat depression and relevant plasma or CSF concentration are much lower than the concentrations used in this study.
  • In Figure 4, the TEA concentration used is noted to be 25 mM, a value which is virtually quite high. In lines 168-169, the statement seems overestimated and needs to be modified
  • In lines 242-244, the statement regarding calcium ion channels is unclear and needs to be modified. Ketamine effect of sodium current needs to be stated as well.
  • In lines 254-260, please revised the paragraph significantly. Please consider quoting the paper (Huang et al., Neurotoxicology, 2012;33:1058-1066) in the revised manuscript. As described in lines 32-39, there indeed are a wide variety of ionic currents where calcium-activated potassium currents need to be included, which could be adjusted by ketamine treatment.
  • In lines 323-325, please check if the recording electrode was filled with ACSF, rather with K+ containing solution. Did the investigators perform the experiments with extracellular recordings or with whole-cell patch-clamp recordings? In line 323-324, two-step vertical puller (PP-830) was noticeably used in the study. Please explicitly describe electrophysiological measurements used in the work.
  • The objective or rationale of the presence study appears to be unclear. The Introduction section of the manuscript hence needs to be modified. The Discussion section needs to be revised for the interpretation of the current experimental findings.

Author Response

We thank the Reviewer for their helpful comments. We have adopted all of their ideas in our revised manuscript (please see “Track Changes” for changes). Below, we have list point-by-point our responses to the Reviewer's suggestions:

1 - In Figures 1A, 1B, 1C and 1D, the concentration of ketamine was noticeably used at 10 and 350 microM. However, the PS amplitude (% of control) appeared no difference between these two concentrations. Please show the “dose-dependent” effect on changes in PS amplitudes.

The reviewer is correct; there were no differences seen between the effects of the two concentrations. We have added the following statement to our manuscript: “Comparing the two concentrations studied, there were no differences seen in the magnitudes of post-washout increases in excitability.” (Lines 77-78)

2 - In Figures 1C, D and Figure 2 (and lines 331-332), the concentration of ketamine used is 350 microM, a value which is surprisingly higher and difficult to be reached in the hippocampal region occurring in vivo. In lines 290-292, the statement was hence over-stated. Is there any pharmacokinetic studies which can be quoted or cited for this study? The typical antidepressant dosage of ketamine used to treat depression and relevant plasma or CSF concentration are much lower than the concentrations used in this study.

As requested by the Reviewer, we did reference our concentration values for both the high and low concentrations of ketamine that we used; please see Lines 59-60. We have also added the following statement to our manuscript: “New research suggests that antidepressant effects occur at both low concentrations of ketamine for the clinical treatment of depression and high concentrations of ketamine used during general anesthesia (personal communication from Boris D Heifets, based on his new clinical trial results).” (Lines 309-312)

3 - In Figure 4, the TEA concentration used is noted to be 25 mM, a value which is virtually quite high. In lines 168-169, the statement seems overestimated and needs to be modified

As requested we provide a citation for the TEA concentration used: “(25 mM: [28]).” (Line 173).

4 - In lines 242-244, the statement regarding calcium ion channels is unclear and needs to be modified. Ketamine effect of sodium current needs to be stated as well.

The Reviewer is correct.  We have modified the statement: “…which includes NMDA receptors, G-protein-coupling, and calcium ion channels.” (Lines 257-258)

5 - In lines 254-260, please revised the paragraph significantly. Please consider quoting the paper (Huang et al., Neurotoxicology, 2012;33:1058-1066) in the revised manuscript. As described in lines 32-39, there indeed are a wide variety of ionic currents where calcium-activated potassium currents need to be included, which could be adjusted by ketamine treatment.

We thank the Reviewer for pointing out the ketamine effect on calcium-activated potassium currents. We have added the suggested reference and have included the following statement: “…and replicates previous findings showing that ketamine blocks calcium-activated potassium currents [36].” (Lines 271-272)

6 - In lines 323-325, please check if the recording electrode was filled with ACSF, rather with K+ containing solution. Did the investigators perform the experiments with extracellular recordings or with whole-cell patch-clamp recordings? In line 323-324, two-step vertical puller (PP-830) was noticeably used in the study. Please explicitly describe electrophysiological measurements used in the work.

Thank you for helping us clarify our recording methods. We have added the following statement: “Extracellular recordings of PS and fEPSP responses were used throughout this study.” (Lines 348-349)

7 - The objective or rationale of the presence study appears to be unclear. The Introduction section of the manuscript hence needs to be modified. The Discussion section needs to be revised for the interpretation of the current experimental findings.

Thank you for helping us clarify the rationale. We have added the following statement to our introduction: “The rationale for this study was to determine whether NMDA receptors were critical to ketamine’s action and whether long-lasting effects were produced.” (Lines 52-53)

We have also added the following statement to our discussion section to clarify our interpretation of the experimental findings: “We interpret our results as evidence that ketamine acts downstream from NMDA receptors to produce a long-lasting increase in CA1 neuron excitability that could contribute to its antidepressant effects by shifting the excitation/inhibition coupling towards a more excitable state.” (Lines 312-315)

Reviewer 2 Report

some abreviation need to be defined NMDA  is well known but others need an initial explanation 

introduction 

Line 28 , (recently)  please provide a recent reference as the reference is year 2000 which is 21 y ago or "delete recently".

a figure locating different receptors would be useful as in figure 6

 Figure 2 is  the difference between amplitude  of the spikes is significant ? 

Line 107 Please elaborate if possible by additional reference 

Line 115 please rephrase and elaborate , this is not very clear to me 

discussion 

please provide reference  line 265

please provide reference  line  282

line 288 -292 , please rephrase and delete ..  at the end and provide reference 

Please explain  how do you differentiate postsynaptic effect of ketamine this is not very clear to me 

please make a junction of this experiment to real life clinical situation 

Please write a conclusion at the end of the discussion resuming clearly what the experiment achieved and suggest future research 

methods 

brain slice preparation and electrophysiological preparation  , please insert a reference using the same methods;

Please explain why there is no any ketamin group without any  washout ? or at least a wash out begining 2 h later only 

Author Response

We thank the Reviewer for their helpful comments. We have adopted all of their ideas in our revised manuscript (please see “Track Changes” for changes). Below, we have list point-by-point our responses to the Reviewer's suggestions:

1 - Line 28 , (recently)  please provide a recent reference as the reference is year 2000 which is 21 y ago or "delete recently".

The Reviewer is correct; 2000 is not very recent. We have deleted the word ‘Recently’ from Line 28.

2 - a figure locating different receptors would be useful as in figure 6

Unfortunately, the locations of receptors and ion channels are not very well determined and appear to be wide-spread on pyramidal cells, interneurons and even glial cells in the hippocampal CA1 area. We cannot provide a figure, as requested, for this reason.

3 - Figure 2 is  the difference between amplitude  of the spikes is significant ? 

We thank the reviewer for posing this question. Significant differences were evident for ketamine’s acute effect, for the bicuculline reversal, and for the post washout increase in amplitude. Please see also Figure 5.

4 - Line 107 Please elaborate if possible by additional reference 

We have added the following references [4,5] to Line 109.

5 - Line 115 please rephrase and elaborate , this is not very clear to me 

We thank the Reviewer for allowing us to clarify. We have added the following statement: “If the LTP-like effect was produced by conventional means, we would expect to see an increase in fEPSP amplitudes following washout of ketamine, which would signify an increase in synaptic transmission to be causing this excitatory effect.” (Lines 115-118).

6 - please provide reference  line 265

We have added the following reference [6] to Line 276 to explain the preceding sentence.

7 - please provide reference  line  289

We have added the following reference [39] to Line 292.

8 - line 288 -292 , please rephrase and delete ..  at the end and provide reference 

We have modified the following sentence “…provides a plausible mechanism for ketamine’s antidepressant effects in vivo,” and have included the following reference [40] in Lines 298-299 for clarity. We have also added onto the last paragraph to provide further reference as well as our interpretation of the results from this study (Lines 309-315).

9 - Please explain  how do you differentiate postsynaptic effect of ketamine this is not very clear to me 

Thank you for allowing us to clarify this. We have added the following statement to our manuscript to explain: “We differentiate postsynaptic effects from presynaptic effects by comparing ketamine’s action on fEPSPs with those on CA1 neuron population spikes (PS). We saw no change in fEPSPs when ketamine was applied nor was there any evidence of a post-washout increase; this can be interpreted as little to no effect at the synapse as shown in Figure 3A. We also performed an input-output analysis (Figure 3C) which showed that PS amplitudes increased with little to no change in fEPSP responses. Thus, ketamine is acting downstream from the synapse to increase the discharge of CA1 neurons (PS).” (Lines 246-253)

10 - please make a junction of this experiment to real life clinical situation 

We have added the following reference to link increased excitability to antidepressant effects in vivo: [40]. (Line 299)

11 - Please write a conclusion at the end of the discussion resuming clearly what the experiment achieved and suggest future research 

Thank you for this helpful suggestion.  We have added the following statement to our conclusion to clarify our results and suggest directions for future research: “New research suggests that antidepressant effects occur at both low concentrations of ketamine for the clinical treatment of depression and high concentrations of ketamine used during general anesthesia (personal communication from Boris D Heifets, based on his new clinical trial results). We interpret our results as evidence that ketamine acts downstream from NMDA receptors to produce a long-lasting increase in CA1 neuron excitability that could contribute to its antidepressant effects by shifting the excitation/inhibition coupling towards a more excitable state.” (Lines 309-315)

12 - brain slice preparation and electrophysiological preparation  , please insert a reference using the same methods;

Thank you, we already included this is in our original manuscript. Please see Lines 320-321: “Methods for brain slice preparation and electrophysiological recordings have been described elsewhere [30], but we include a cursory description here.”

13 - Please explain why there is no any ketamin group without any  washout ? or at least a wash out begining 2 h later only 

This is a great suggestion, and we will incorporate these groups in future experiments.

Round 2

Reviewer 1 Report

Question 2:
Please quote the published references to demonstrate that the ketamine concentration greater than 300 microM is therapeutically relevant.

Question 6:
Please elaborate extracellular recordings in detail. For example, what kind of puller was used in the study for extracellular recordings? What was the recording solution filled in the pipette used in this work? Please describe in the Materials and Methods.  How would the white noise be removed in the study, since some of experimental observations appear to be unclear..

Since most of cellular electrophysiological recordings have been made, please state the novelty in this study.

Author Response

Response to Reviewers:

Reviewer 1:

Question 2:
Please quote the published references to demonstrate that the ketamine concentration greater than 300 microM is therapeutically relevant.

As pointed out previously, ref 22 provides this information: “representing each group at both low and high concentrations (1 or 4 MAC for isoflurane [Iso] and halothane [Halo]; 10 or 500 µM for ketamine [Ket]; 10 or 100 µM for propofol [Prop])”

See also:

Ketamine destabilizes growth of dendritic spines in developing hippocampal neurons in vitro via a Rho-dependent mechanism.

Jiang S, Hao Z, Li X, Bo L, Zhang R, Wang Y, Duan X, Kang R, Huang L. Mol Med Rep. 2018

“hippocampal neurons were exposed to different concentrations (100, 300 and 500 µM) of ketamine”

We also note in our discussion: “Other limitations include the lack of information regarding the effect site concentration of ketamine in human subjects from clinical research studies as well as the unknown concentration of ketamine within hippocampal slices at the site of action in our experiments [42].”

Question 6:
Please elaborate extracellular recordings in detail. For example, what kind of puller was used in the study for extracellular recordings? What was the recording solution filled in the pipette used in this work? Please describe in the Materials and Methods.

Please see the following in the original text:

“Recording electrodes were made from glass micropipettes (Garner Glass, Claremont, CA) that were pulled with a two-stage pipette puller (Model PP-830, Narishige, Japan) to 2-5 MOhm tip resistance, when filled with ACSF. Recording electrodes were placed near the CA1 cell body layer in the stratum oriens for PS responses or in the stratum radiatum for fEPSP responses. Extracellular recordings of PS and fEPSP responses were used throughout this study.”

How would the white noise be removed in the study, since some of experimental observations appear to be unclear.

There was no white noise in our experiments, so it was not removed.  All observations shown in our figures are very clear.

Since most of cellular electrophysiological recordings have been made, please state the novelty in this study.

Please see our original text:  “This is this first report of a long-lasting increase in excitability following ketamine exposure.”

Reviewer 2 Report

The authors have improved the manuscript and I thank them

I have a minor comment 

If figure cannot be presented please provide minimal explanation in the text such as the receptors are not very well determined etc..

Author Response

Response to Reviewers:

Reviewer 2:

If figure cannot be presented please provide minimal explanation in the text such as the receptors are not very well determined etc..

The following was stated during the last revision: “Unfortunately, the locations of receptors and ion channels are not very well determined and appear to be wide-spread on pyramidal cells, interneurons and even glial cells in the hippocampal CA1 area. We cannot provide a figure, as requested, for this reason.”

This is general knowledge and also would not add new knowledge, so there is no need to add such a statement to our paper.

Round 3

Reviewer 1 Report

The reply from the authors is acceptable.